# Correlative Confocal Raman and Scanning Probe Microscopy in the Ionically Active Particles of LiMn_2_O_4_ Cathodes

**DOI:** 10.3390/ma12091416

**Published:** 2019-04-30

**Authors:** Denis Alikin, Boris Slautin, Alexander Abramov, Daniele Rosato, Vladimir Shur, Alexander Tselev, Andrei Kholkin

**Affiliations:** 1School of Natural Sciences and Mathematics, Ural Federal University, Ekaterinburg 620000, Russia; boris.slautin@urfu.ru (B.S.); alexander.abramov@urfu.ru (A.A.); vladimir.shur@urfu.ru (V.S.); kholkin@urfu.ru (A.K.); 2Robert Bosch GmbH, 70839 Gerlingen-Schillerhoehe, Germany; daniele.rosato@seg-automotive.com; 3Department of Physics & CICECO—Aveiro Institute of Materials, University of Aveiro, 3810–193 Aveiro, Portugal; atselev@ua.pt (A.T.); kholkin@ua.pt (A.K.)

**Keywords:** low-frequency electrochemical strain microscopy, confocal Raman microscopy, scanning probe microscopy, lithium manganate, LiMn_2_O_4_, quantitative electrochemical strain microscopy, batteries, cathodes

## Abstract

In this contribution, a correlative confocal Raman and scanning probe microscopy approach was implemented to find a relation between the composition, lithiation state, and functional electrochemical response in individual micro-scale particles of a LiMn_2_O_4_ spinel in a commercial Li battery cathode. Electrochemical strain microscopy (ESM) was implemented both at a low-frequency (3.5 kHz) and in a high-frequency range of excitation (above 400 kHz). It was shown that the high-frequency ESM has a significant cross-talk with topography due to a tip-sample electrostatic interaction, while the low-frequency ESM yields a response correlated with distributions of Li ions and electrochemically inactive phases revealed by the confocal Raman microscopy. Parasitic contributions into the electromechanical response from the local Joule heating and flexoelectric effect were considered as well and found to be negligible. It was concluded that the low-frequency ESM response directly corresponds to the confocal Raman microscopy data. The analysis implemented in this work is an important step towards the quantitative measurement of diffusion coefficients and ion concentration via strain-based scanning probe microscopy methods in a wide range of ionically active materials.

## 1. Introduction

Li-ion accumulators (LIA) are one of the important factors for the development of systems for energy storage, transport, and mobile electronics. The worldwide production of LIA is rapidly growing [1,2]. Functional investigations of the LIA materials by microscopic methods play a key role for understanding of the LIA due to their ability to study electrochemical processes occurring in batteries at the micro- and nanoscale. Local studies of the materials for the electrochemical power sources using scanning probe microscopy (SPM) started simultaneously with the appearance of the first commercial scanning probe microscopes [3]. At the beginning, topography change under the macroscopic actions (cycling, intercalation, deintercalation, etc.) and local action of the voltage applied to the atomic force microscopy (AFM) tip were used by researchers [3,4]. Later, methods allowing inspection of ionic conductivity at the nanoscale, so-called scanning electrochemical microscopy (SECM) [3,5], were developed, which are based on direct measurements of ion interactions with a sample in an electrolyte media across the sample area. However, these methods require complicated sample preparation, specialized cells and probes, as well as a high homogeneity of the surface, and need to be implemented in a liquid electrolyte media. This makes their usage complicated for implementation in many practical electrochemical systems [5].

Balke with co-authors suggested an alternative approach, so-called Electrochemical Strain Microscopy (ESM) [6,7], based on registration of local strain in a material appearing as a result of application of a DC and/or AC voltage with conventional silicon AFM probes with a conductive coating [8]. The strain is believed to be caused by the Vegard expansion induced by the local change of the ion concentration [9,10]. The corresponding response was observed in a wide range of ionically active materials, such as lithium battery materials: lithium manganites [11] and cobaltites [7,12], amorphous silicon (used as an anode material in lithium-ion batteries) [6], nanocrystalline LiFePO_4_ [13], a high-temperature ionic conductor cerium oxide in the form of thin films [14,15] and bulk ceramics [16], lithium manganate extracted from commercial lithium batteries [17,18], solid state electrolytes [19], and even organic transistors [20]. A special case proving the possibility to induce large rearrangements of ions under the action of the tip-induced electric field is the formation of metal islands on the surface of some Li- and Ag-conducting glasses [21,22].

Nevertheless, in most of the cases, the nature of the ESM response is still under discussion because of a variety of other parasitic processes accompanying the electromechanical strain in the voltage modulated AFM measurements. Most of the parasitic contribution are associated with the application of a DC voltage, such as charge injection as well as local electrochemical reactions and phase transitions [18,23]. Recently, Seol et al. demonstrated an alternative approach based on implementation of the ESM without application of a DC bias in a low-frequency AC range in Li-ion conductive glass ceramics [24]. Later, our group developed a theoretical basis for a quantitative ESM in mixed ionic-electronic conductors and quantitatively evaluated local diffusion coefficients in commercial battery cathodes [25].

However, some parasitic contributions can take place even in the case when only an AC voltage is used in the measurements. For example, the local electrostatic force was found to play a dominating role in the ESM in the on-resonance ESM measurements [26]. Another possibility is the local Joule heating of a sample occurring in a mixed electronic-ionic conductor with intense electronic current flows [27,28]. Moreover, an ESM response can occur together with different physical phenomena taking place simultaneously. For example, an AFM probe can respond to an electrostatic force modulated by ionic motion in a manner similar to that observed in the electrostatic force microscopy in Li ionic conductors [29], or variations of the ion concentration can be induced by local heating, such as in the scanning thermo-ionic microscopy [28].

Thus, the knowledge of the local ionic and electronic transport is incomplete without comprehensive studies of samples with methods of the microscopic local structural analysis together with the scanning probe microscopy. The present work fills this gap by employing correlated ESM, conductive atomic force microscopy (C-AFM), and confocal Raman microscopy, which allows obtaining complementary images of the structural changes, phase composition, and degree of lithiathion/delithiation together with functional properties, such as electronic conductivity and electrochemical strain response. Based on this approach, different parasitic responses in the scanning probe measurements were systematically separated and analyzed. Correlations between material states, such as parasitic phases and Li ion concentration in LiMn_2_O_4_ cathodes, were evaluated from the ESM data.

## 2. Materials and Methods

A Li-battery cathode (positive electrode) was investigated in our experiments. The cathode sample consisted of Li_0.61_Mn_2_O_4_ (LMO) active particles embedded in a carbon binder mixed with poly(vinylidene fluoride) (PVDF). The particles have a cubic spinel lattice structure with a nominal lattice parameter of 0.809 nm as was determined from X-ray diffraction measurements of identical samples in [30,31]. The average concentration of Li ions can be estimated from these data to be about 1.15 × 10^27^ m^−3^. The LMO cathode was prepared and tested in an electrochemical cylindrical cell at Robert Bosch GmbH (Gerlingen, Germany). It was cycled against graphite anodes at a 1 C-rate from 2.5 V to 4.2 V three times with an 1 C charge rate and 1 C discharge rate. The cells were opened in an argon-filled glove box. The cathode, separator, and anode were uncoiled and separated from each other. Pieces of the cathode on the current collector were cut and washed in a dimethyl carbonate solution in order to remove the electrolyte. After that, samples were taken out of the glove box and dried. Then, cathodes were thoroughly polished by step-by-step decreasing the polishing abrasive grit size with a finish by the stress-free silica fine polishing. As the last step, the sample surface was treated by Ar plasma to thin and possibly remove the damaged surface layer. A more detailed procedure of sample preparation can be found elsewhere [17,18].

Confocal Raman microscopy was carried out using a confocal Raman microscope Alpha 300 AR (WiTec GmbH, Ulm, Germany). A 488 nm-wavelength solid state laser together with an 1800 lines-per-mm diffraction grating was used for the spectroscopic measurements. The laser beam was focused on a cathode particle into a 260 nm-diameter laser spot using a 100×, 0.75 NA objective. Control of the particle heating under the focused laser light was done by carefully controlling the Raman spectra stability. The Raman signal was collected at each scanning point for 25 s. The obtained spectra were fitted by superpositions of Lorentzian functions. Positions and intensity of individual peaks determined after the fitting were plotted as pseudo-color maps with the help of the Gwyddion software (Department of Nanometrology, 2.53, Brno, Czech Republic) following a procedure described in our earlier publication [32].

SPM was carried out with an MFP-3D (Oxford Instruments, Abingdon, UK) and an NTEGRA Aura (NT-MDT Spectral Instruments, Zelenograd, Russia) scanning probe microscopes. ESM measurements were performed with the MFP-3D SPM in a closed cell under flow of dry nitrogen. HR_NC (37 N/m spring constant), HA_NC, (11 N/m spring constant) and HA_FM (3.5 N/m spring constant) W_2_C-coated probes (ScanSens GmbH, Hamburg, Germany) were used for the measurements. The ESM measurements were made in low- and high-frequency ranges. The low-frequency ESM measurements were carried out at a frequency of 3.5 kHz in the single-frequency mode. The high-frequency ESM was conducted at a frequency of the first flexural resonance mode of the cantilever probe (400–2000 kHz) to utilize the resonance amplification. The resonance tracking was accomplished via implementing the dual AC resonance tracking (DART) capability of the MFP-3D microscope [33]. A 5 kHz to 20 kHz-wide frequency window was used in the DART measurements with an AC voltage amplitude from 10 V to 20 V applied to the SPM probe. A 1 ms to 30 ms time constant was used to optimize the scan time and signal-to-noise ratio.

Probe tip displacements in the electromechanical (EM) SPM measurements were determined following the method described in [34,35]. Calibration of the probe tip displacements was made based on the quasi-static force-distance curves. The actual on-resonance values of the cantilever bending in the DART mode were determined from the measured amplitudes following Gannepalli et al. [33]. The obtained values were divided by the corresponding quality factor of the contact resonance and the resonance mode shape factor [34]. The shape factor and contact stiffness were calculated using a home-made MATLAB program (2018a, MathWorks, Natick, MA, USA) based on the measured contact and free resonance frequency values according to procedure described in [34,35]. Parameters of the probes as well as determined contact stiffness and shape factors are summarized in Appendix A (Table A1). The resonance frequency of the first flexural contact mode for the 3.5 N/m probe was determined using the ESM response, while for the 11 N/m and 37 N/m probes, auxiliary piezo-transducer-induced sample vibrations were used to excite the probe oscillations and to measure the contact resonance peaks.

C-AFM measurements were performed on the NTEGRA Aura SPM under vacuum at a pressure of about 10^−3^ Torr. HA_NC W_2_C coated cantilevers (Scansens) with a spring constant of about 11 N/m were used for the measurements. Local current-voltage (I−V) curves were acquired by application of triangle-shape voltage waves with an 8 V amplitude and a 2 s period. The I−V curve measurements were carried out over a matrix of points across the active particle surface, and the C-AFM maps were constructed using values of current at a selected positive or negative bias. Fitting of the I−V curves was made with a home-made Python program. The parameters of the fittings were extracted and plotted as pseudo-color maps with the help of the Gwyddion software.

Surface potential measurements were made with the MFP-3D SPM using the amplitude-modulated Kelvin probe force microscopy (KPFM) mode with a 30 nm tip-surface distance and a 0.5 V AC voltage applied to the tip. 

## 3. Results and Discussion

### 3.1. Confocal Raman Microscopy

As the first step, a number of LiMn_2_O_4_ particles were inspected with complimentary AFM and confocal Raman microscopy. The measurements were conducted at different spots across the cathode sample. An AFM topography image of a typical particle is presented in Figure 1a. According to our previous studies, a few peaks of the Raman spectrum can be used to determine the lithiathion state and phase composition of LMO [32]. At a nominal composition Li_0.61_Mn_2_O_4_, Raman spectra consist of 5 peaks: 310 cm^−1^, 490 cm^−1^, and 630 cm^−1^ related to the F_2_ lattice vibrational modes and peaks at 565 cm^−1^ and 592 cm^−1^ related to the A_1_ modes [32]. Details of the Raman spectra are discussed in [32]. Here, to construct confocal Raman maps, the A_1_ peak at 592 ± 4 cm^−1^, which indicates the local lithiation state, was used (Figure 1b) as well as the peak at 660 ± 3 cm^−1^ associated with the A_1g_ mode of Mn_3_O_4_ (Figure 1b). A higher value of the Raman shift of the A_1_ peak of LMO corresponds to a higher lithiation degree. From the map in Figure 1c, it is evident that the lithiation degree gradually increases towards the grain boundaries, while the center of the particle is slightly less lithiated. At the same time, the strong A_1g_ peak at 660 cm^−1^ is indicative of the Mn_3_O_4_ phase. The Mn_3_O_4_ phase was observed mainly in the vicinity of the particle-binder interface, where it forms distinct clusters (Figure 1d). This behavior is expected due to the Mn dissolution in the electrolyte during cycling (the cathodes were cycled 3 times before delithiation) and supposed to be a key factor of the electrode capacity degradation [36].

### 3.2. Low-Frequency Electrochemical Strain Microscopy

To inspect distributions of the Li-ion concentration and diffusivity, we used the low-frequency ESM (LF ESM). The 3.5 kHz ESM images in Figure 2a–c reveal variations of both amplitude and phase of the EM response across the particle. As seen in Figure 2b, the amplitude varies from about 12 pm to 14 pm with *V_ac_* = 10 V. The EM response in the Li-ion conductors that are nominally not EM-active, like LMO, is attributed to the local alterations of the Li ion concentration [7,37]. An increase of ion concentration in the vicinity of an SPM tip results in a local volume expansion, the so-called Vegard strain [7]. Application of a DC voltage to the SPM tip leads to a local Li electro-migration with Li intercalation or deintercalation, depending on the polarity of the applied voltage, while an AC voltage induces a reciprocating electro-migration of Li ions [25]. A large enough DC electric field causes a topographic change, while an AC voltage induces surface vibration at the corresponding frequency. The amplitude of the AC-induced surface displacement usually lies in the pm range, which is within the range of the SPM vertical resolution [25,38].

Nevertheless, the level of the EM response during the LF ESM measurements is significantly lower than a typical piezoresponse signal from a piezo-electric [39,40]. Therefore, it can be comparable to parasitic signals of different origins [40]. Hence, before analysis of the LF ESM maps, we estimate possible contributions from parasitic effects. 

The strongest parasitic contributions discussed in literature are the converse flexoelectric effect [41], electrostatic tip-surface interaction [42], and local Joule heating in the contact area [27]. The converse flexoelectric effect induces surface displacements existing in every dielectric material due to polarization in a stress gradient [41]. In our experiments, the measured ESM response was found to be independent of the force applied to the probe, and, based on the analysis in [34], the contribution due to the flexoelectric effect can be excluded. Further, our samples are mixed ionic-electronic conductors, and, thus, the tip-surface interface is non-blocking for transport of electrons and blocking for ions. Hence, both the electrostatic force and local Joule heating are expected to contribute to the electromechanical response. We discuss them in detail below.

### 3.3. Parasitic Contributions to the ESM Response

#### 3.3.1. Electrostatic Tip-Surface Interaction

The electrostatic tip-surface interaction associated with a non-zero sample surface potential results in a tip response at the frequency of the applied AC voltage, which can be expressed as [42]: (1)$$S_{\omega }=\frac{1}{k^{*}}C^{\prime }\cdot U_{sp}U_{ac},$$
where *k** is the tip-sample contact stiffness, *C*′ is a coefficient depending on the tip-sample system geometry and electrical properties of the sample, and *U_sp_* is a surface potential. We note that the response due to this mechanism is frequency-independent. 

In order to clarify the role of the electrostatic force in the observed ESM contrast, we performed ESM measurements at a higher frequency. Since the electrostatic response is expected to be independent of frequency while the response due the Vegard expansion drops with increasing frequency as *f*^−1/3^ [25], the increase of the measurement frequency effectively eliminates the ESM contrast from the images. To further increase the signal from the electrostatic tip-surface interaction, experiments can be performed at a smaller tip-sample contact stiffness *k** (Equation (1)).

The high-frequency ESM was carried out in the DART mode where resonance amplification is used for sensitivity enhancement. It must be noted that no EM signal was found in the high-frequency range with the 37 N/m-stiff cantilever (3890 N/m contact stiffness). Some measurable response can be observed only with softer cantilevers with a 11 N/m spring constant (910 N/m contact stiffness) as displayed in Figure 2d–f. As seen, application of an AC voltage as high as 10 V in amplitude at a frequency of about 1 MHz does not result in any significant response (Figure 2e). Indeed, the corresponding tip displacement amplitude was below 0.01 pm/V. Thus, we further inspected the sample with an even softer probe with a spring constant of 3.5 N/m at a 620 N/m contact stiffness (Figure 2g–i). The EM measurements with the softer probe at about 440 kHz clearly show distribution of amplitude and phase across the particle surface (Figure 2g,h). However, the EM image contrast in this experiment is mainly due to a cross-talk with sample topography and significantly differs from that in the low-frequency ESM images. These observations point to different mechanisms of EM response at the low and high frequencies. The EM probe tip displacement measured at the high frequency is about 0.1 pm/V. We performed numerical calculations of the electrostatic force acting onto the AFM tip employing an ‘AC–DC’ module of the COMSOL Multiphysics v.5.3a (COMSOL Inc., Stockholm, Sweden) finite element simulations package (Appendix B, Figure A1). The calculated result was applied to calculate the tip displacements caused by the electrostatic force. The obtained value of 0.1 pm/V is in a close match with the high-frequency experimental result. 

Assuming a dominant electrostatic response with the 3.5 N/m probe, we can determine the strength of the electrostatic tip-surface interaction from Equation (2) and apply the result to the low-frequency experiments with the 37 N/m probe by substitution of the corresponding contact stiffness, *k**, in Equation (2). This estimate yields around 0.01 pm/V, that is, significantly less than the obtained displacement values of ~1.3 pm/V for the 37 N/m-stiff probe and near the technique sensitivity. This clearly points to a negligible contribution from electrostatic force in the LF ESM data in Figure 2a–c.

#### 3.3.2. Local Joule Heating 

Next, we turn to consideration of the Joule heating due to electric current flow in the tip-sample contact area and its contribution to the LF ESM response in our experiments. Local Joule heating in a tip-surface contact leads to a thermal expansion of the sample, which results in corresponding surface displacements. If the tip-sample electric conduction is ohmic, a mechanical response of the AFM probe can be detected only at the second harmonic of the applied AC voltage [27,28]. However, situation becomes different if the I−V characteristic is non-linear and is not perfectly anti-symmetric in respect to the applied voltage. The general equation for the time-dependent electromechanical response to the applied AC voltage due to the Joule heating with a non-linear I−V characteristic can be expressed as:(2)$$S(t)=\beta_{J}P(t)=\beta_{J}I(U(t))\cdot U(t),$$
where *β**_J_* is a Joule heat transduction coefficient, i.e., a factor describing conversion of the Joule heat into the mechanical displacement. Generally, this factor is frequency-dependent. To find the response at the *n*-th harmonic of the applied AC voltage $U(t)=U_{ac}\cdot \mathrm{Cos}(2\pi ft)$, the *n*-th coefficient of the cosine Fourier series expansion of the displacement *S(t)* should be calculated:(3)$$S_{n}=4f\int_{0}^{1/(2f)}S(t)\mathrm{Cos}(2\pi nft)dt.$$

We used I−V curve mapping to evaluate the electronic current across the particle surface (Figure 3). To this aim, I−V curves were acquired at every pixel by moving the probe from pixel to pixel across the particle. A bias voltage in these experiments was applied to the probe. A typical I−V curve is presented in Figure 3b. Current maps obtained at fixed bias voltages of ±3 V DC are presented in Figure 3c,d. It is seen that the current weakly varies across the surface more or less correlating with topography (Figure 3a–c). Some non-topographic features in the image contrast can be associated with redistribution of lithium ions by the electric field of the probe. The obtained I−V curves are non-linear and asymmetric (Figure 3b). The current is usually larger at a positive bias than at the negative bias of the same value (Figure 3c,d). The asymmetry of the I−V curves can be traced to changes of the material conductivity, which is a function of Li-ion concentration [43]. Application of a nominally small bias voltage to an AFM tip induced a quite high local electric field (see Appendix B). The field value is above 10^−7^ V/m, that is, high enough for electron and hole injection into a sample [44]. The injection of additional charge carriers in the material leads to a space charge limited electronic transport (space-charge-limited current, SCLC) [44,45,46].

For the purpose of a simplified analysis of the EM response here, we assume that the I−V curve shape can be described by the SCLC model [44,45,46]:(4)$$I_{sclc}(t)=A^{\pm }\cdot U^{n},$$
where *U* is an applied DC voltage, *n* is a SCLC exponent, which can vary between 2 and 4 depending on the properties of charge traps in the current conducting material, and *A**^±^* are the SCLC coefficients generally assuming different values for positive and negative *U*. From Figure 3b, it is clearly seen that the I−V curve is well fitted by Equation (4), thereby proving applicability of the SCLC model. Generally, values of *n* are different also for the positive and negative bias due to differences in injection of electrons and holes into the material. The difference in *n* was found to be of a minor importance in our experiments (Figure 3b) and was neglected in the subsequent consideration.

The I−V curves obtained at each pixel were fitted by Equation (4) with *A**^±^* and *n* being fitting parameters; the values of *n* were set to be equal for the positive and negative branches of the I−V curves. After that, the values of *A**^±^* and *n* were plotted as separate maps, and the maps are displayed in Figure 3e–g. As seen, the exponent *n* varies between 2 and 4, which is in line with the SCLC model, while the average values of the *A**^±^* coefficients are different by a factor not exceeding 2 for the positive and negative branches (Figure 3h). This fact has an important consequence for the possible heating-induced EM response. Indeed, in the absence of a DC voltage offset in the AC measurements and with a linear I−V dependence, the dissipation of the Joule power, *P*, must result in a temperature oscillation and a corresponding sample expansion, *S*, only at the second harmonic of the applied AC voltage because $S\sim P\sim U_{ac}^{2}\left(Cos(2\pi ft\right){)}^{2}$ (Equation (2)), that is, the first harmonic signal must be zero. However, with a non-linear I−V curve, the situation can be different. An example of the AC-voltage-induced heat generation as a function of time for different values of the parameters *A**^±^* with *n* = 2.5 in Equation (4) are represented in Appendix C (Figure A2). It is clear from the plots that only asymmetric I-V dependences can be a source of a first harmonic EM response. Calculations with use of Equation (3) show that the amplitude of the Joule power oscillations at the first harmonic for our maximal I−V curve asymmetry (*A*^+^/*A*^−^ = 2) is about 50% of that at the second harmonic.

In order to estimate the value of the EM response due to the local heating, we performed FE numerical simulations of the Joule heat generation and thermal expansion with use of ‘Joule heating’ module of the COMSOL Multiphysics v.5.3a finite element simulations package (Appendix D, Figure A3). The approximation of an Ohmic contact was used in the FE model. The frequency of the AC current was half of the experimental one, and the current amplitude and material conductivity were adjusted to produce the first harmonic dissipated power oscillations calculated with Equation (3) (applied to the power) for the maximal asymmetry of the experimental I−V curves. A surface displacement amplitude of about 0.01 pm/V was obtained in the modelling, which is significantly less than the measured LF ESM response.

### 3.4. Relationship between Electrochemical Strain, Composition, and Lithiation State

Summarizing the previous discussion, we conclude that the LF ESM measurements performed by the stiff cantilevers are free from parasitic responses and, therefore, they can be used for qualitative comparison with the confocal Raman microscopy data. The contrast correlation between the images obtained with the two techniques (Figure 1c,d and Figure 2b,c) was found to be imperfect, but this is expected due to a significantly different in-depth sensitivity of the Raman spectroscopy and SPM measurements. The penetration of the laser light is about 300 nm, while the in-depth sensitivity of the SPM is a few tip-sample contact radii, that is <50 nm. Moreover, the penetration of the electric field from the AFM tip is limited due to a voltage drop across the low-conductivity surface layer [25]. Nevertheless, some distinguishable similarity of the images can be found.

Surprisingly, in the area with the segregated Mn_3_O_4_ phase, the ESM signal is larger in comparison with the rest of the particle. In turn, the EM signal outside the particle, i.e., from the binder composed of the PVDF mixed with carbon black is shifted in phase by 180 degrees in comparison with the signal from the active particle (Figure 2c and Figure 4a,b). At the same time, the measurements of the surface potential (not shown) revealed a uniform surface potential distribution inside and outside the particle area with an average value close to –160 mV. Thus, the 180-degree phase shift across the particle-binder interface (Figure 4a,b) cannot be caused by a surface potential change and associated with the electrostatic tip-surface interaction and must be ascribed to a change of the EM response mechanism. We interpret this as a possible indication of a piezoresponse from the β-phase of the PVDF, which has a negative electrostriction coefficient [16].

The ESM phase signal in the area with the segregated Mn_3_O_4_ phase is slightly noisier (compare the areas indicated in Figure 4a by circles). This fact is validated by different widths of the signal histograms displayed in Figure 4c for the indicated areas of the image. To interpret this observation, we note that Mn_3_O_4_ is expected to have a lower Li diffusion coefficient in comparison to LMO [36]. However, a careful analysis of the Raman spectra allows us to identify the LMO peaks even in the areas with the Mn_3_O_4_ segregation (Figure 1b). The corresponding shift of the A_1g_ peak around 660 cm^−1^ in clusters containing Mn_3_O_4_ was 1–2 cm^−1^ greater, which may be associated with an enhanced lithiation degree of LMO in these areas. Thus, enhancement of the ESM signal in the area with the Mn_3_O_4_ phase can be attributed to the presence of the LMO phase with a higher lithiation. The enhancement of the noise is, therefore, due to mixing of electrochemically active and non-active phases.

The ESM signal inside the active particle area in Figure 2b is distributed non-uniformly even outside the areas with the Mn_3_O_4_ phase. These areas are spatially correlated with the distribution of the A_1_ Raman line shift in Figure 1c, and, therefore, can be attributed to a variation of the Li ion concentration over the particle area. For a more detailed picture, we turn to results of our previous publication on the LF ESM of LMO cathodes [25]. Assuming that the charge neutrality holds across the whole sample volume in a mixed ionic-electronic conductor, it was derived in [25] that the surface displacement amplitude, *S_ω_*, under an SPM tip can be calculated as:(5)$$S_{\omega }\approx \frac{1+\nu }{\pi }\beta \left(1-\gamma \right){С}_{0}R_{0}\left(\sqrt[{3}]{1+3\mu_{i}U_{eff}\gamma \frac{1}{R_{0}^{2}f}}-1\right),$$
where *v* is the Poisson ratio, *β* is the Vegard coefficient, $\mu_{i}=Dq_{i}/k_{B}T$ is the ionic mobility, *C*_0_ is Li ion concentrations in the particle, *γ* is a coefficient accounting for the maximal change of the ion concentration due to the tip-induced electro-migration (γ≈1/3 according to [25]), *D* is the diffusion coefficient, *q_i_* is the ionic charge, *R*_0_ is the tip radius, and *U_eff_* is an effective voltage applied to the material bulk. Due to a voltage drop in the surface layer, *U_eff_ ≈ 1/4 U_ac_*, where *U_ac_* is the amplitude of the AC voltage applied between the probe and the sample bottom electrode [25]. In the approximation of a uniform diffusion coefficient across the particle, Equation (5) allows evaluating the local ion concentration, *C*_0_, at each image pixel over the particle. For use with Equation (5) in our setup, *β* ≈ 10^−30^ m^3^ [47], *v* ≈ 0.3 [47], *D* ≈ 10^−9^ cm^2^/s [48], *U_ac_* = 10 V, *R*_0_ = 10 nm, and *f* = 3.5 kHz. The average tip displacement amplitude obtained from the LF ESM images in Figure 2b and Figure 4b is *S*_ω_ = 1.3 ± 0.02 pm/V, which is in agreement with the value 1−2 pm/V expected for the Vegard strain in LMO under our experimental conditions according to measurements and analysis in [25]. The local concentration across the particle area varies from 0.7 × 10^27^ m^−3^ to 1.4 × 10^27^ m^−3^, which, in turn, agrees well with the average concentration value of 1.15 × 10^27^ m^−3^ that follows from the data of the macroscopic XRD measurements of the sample.

For a further analysis, we plotted corresponding line profiles of the A_1_ Raman peak shift and average LF ESM response amplitude as displayed in Figure 5a. The data used to plot Figure 5a were taken along the lines indicated in Figure 1c and Figure 2b. The plot demonstrates a correlation of the two signals, with an enhanced lithiation towards particle boundaries. In Figure 5b, we further plot the Raman shift versus LF ESM response amplitude for a number of corresponding line profiles similar to the profiles in Figure 5a. The degree of the correlation was quantified by calculating a Pearson correlation coefficient and linear fit of the data (Figure 5b). The value of the Pearson coefficient for the data in Figure 5b is 0.8, which indicates a strong correlation.

## 4. Conclusions

In conclusion, we implemented a correlative microscopy approach to the active ionic particles of a lithium manganese spinel inside a commercial battery cathode. It was shown that the low-frequency electrochemical strain microscopy reveals contrast significantly different from that obtained in the high-frequency range. It was shown both experimentally and by finite element modeling that this contrast cannot be attributed to the converse flexoelectric effect, influence of the local electrostatic forces, or electromechanical cantilever oscillations induced by local Joule heating of the sample. The LF ESM signal distribution was found to be correlated with the confocal Raman microscopy data, which demonstrates a possibility of a correlative approach to locally resolve electrochemically-inactive phases and state of charge variations. However, it must be noted that the tip-induced electrochemical strain in lithium manganate results in quite small surface displacements around 1–2 pm per Volt of AC voltage applied to the surface. This value is close to the sensitivity limit of the modern scanning probe microscopes. Reliable ESM signals can be collected only via implementing time-consuming lock-in measurements. One order of magnitude enhancement in the signal-to-noise ratio would be crucial to obtain clearer and more useful signals in the ESM measurements. This can be achieved, for instance, by improvement of the excitation and registration schemes of the cantilever deflection sensor in scanning probe microscopes [49]. Until more sensitive measurements are available, the ESM needs to be supplemented by data obtained with alternative methods.

## Figures and Tables

**Figure 1 materials-12-01416-f001:**
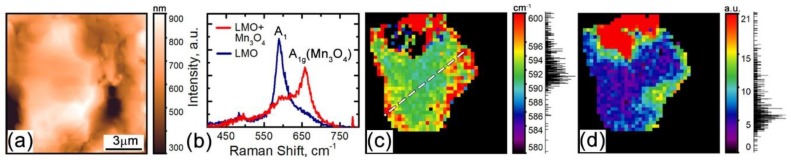
Complimentary atomic force microscopy (AFM) and micro-Raman spectroscopy of an active ionic particle: (**a**) AFM topography image of the area mapped with the micro-Raman spectroscopy. (**b**) Averaged Raman spectra in the Li_0.6_Mn_2_O_4_ and Mn_3_O_4_ phases from the blue and red regions, respectively, of the map in panel (**d**). (**c**) Distribution of the A_1_ Raman peak shift corresponding to the local lithiation state. (**d**) Distribution of the A_1g_ peak intensity reflecting aggregation of the Mn_3_O_4_ phase. Signal histograms are displayed on the right from the color scalebars of the corresponding images. The dashed line in (**c**) indicates the line used to extract the profile of the Raman peak shift displayed in Figure 5a.

**Figure 2 materials-12-01416-f002:**
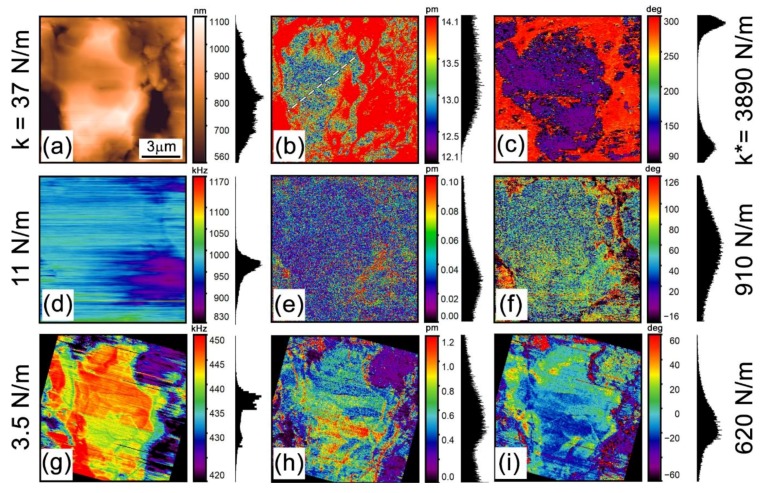
ESM in the active ionic particle shown in Figure 1. Three cantilever probes with different stiffness (denoted on the left side of the figure) were used to obtain the images. The corresponding contact stiffness is shown on the right side of the figure. Images in one row were obtained simultaneously with one probe. (**a**) Topography, (**b**) amplitude, and (**c**) phase measured with the 37 N/m cantilever (*V_ac_* = 10 V) in the 3.5 kHz LF ESM. (**d**) Dual AC resonance tracking (DART) resonance frequency, (**e**) amplitude, and (**f**) phase measured with the 11 N/m cantilever (*V_ac_* = 20 V). (**g**) DART resonance frequency, (**h**) amplitude, and (**i**) phase measured with the 3.5 N/m cantilever (*V_ac_* = 10 V). The signal histograms are displayed on the right from the color scalebars of the corresponding images. The dashed line in (**b**) indicates the line used to extract the profile of the ESM amplitude displayed in Figure 5a.

**Figure 3 materials-12-01416-f003:**
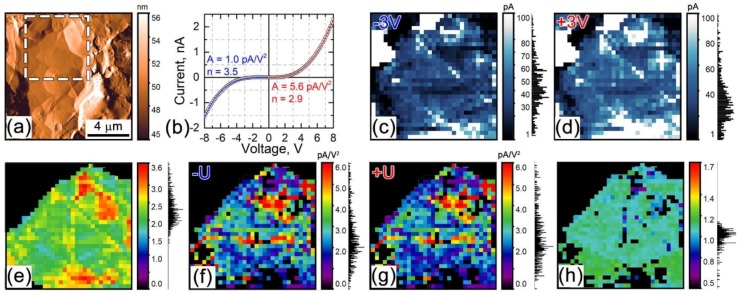
Analysis of local I−V curves and electric current distribution across the active ionic particle. (**a**) Differential topography (deflection error signal). The dashed line shows the 5.5 μm × 5.5 μm area where the I−V curve mapping was performed. (**b**) Typical I−V curve (circles) with a fit by Equation (4) (solid line). (**c**) Current distribution map for V_dc_ = −3 V. (**d**) Current distribution map for V_dc_ = +3 V. Maps of parameters *A* and *n* found by fitting local I−V curves by Equation (4): (**e**) *n*, (**f**) A^−^, (**g**) A^+^, (**h**) A^+^/A^−^_._ Black areas in (**e**–**h**) are regions inside the binder that do not show the typical I−V curve shape. For the images in (**c**–**h**), corresponding signal histograms are displayed on the right from the color scalebars.

**Figure 4 materials-12-01416-f004:**
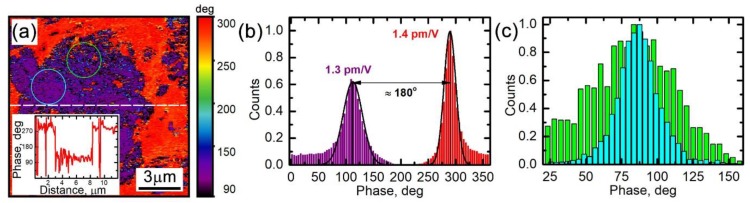
Analysis of the LF ESM signal phase in the active ionic particle. (**a**) LF ESM phase; the inset shows the line profile of the image along the dashed line. (**b**) Histogram of the image in panel (**a**); average EM response amplitudes corresponding to the histogram peaks are indicated near the peaks. (**c**) Local histograms for the signal in the image in (**a**) from the areas marked in (**a**) by the circles with corresponding colors.

**Figure 5 materials-12-01416-f005:**
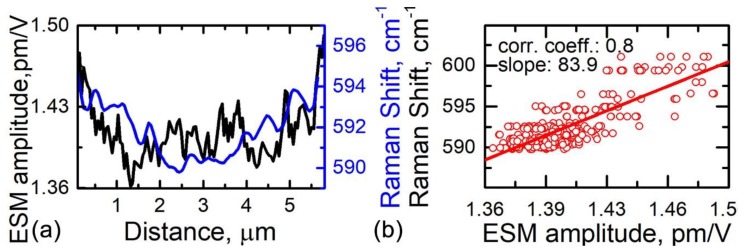
Correlation analysis of confocal Raman microscopy and ESM data. (**a**) Overlapped image profiles along the corresponding lines indicated in Figure 1c (Raman shift) and Figure 2b (ESM amplitude). (**b**) Circles: experimental A_1_ Raman peak shift versus ESM amplitude for a number of line profiles similar to those shown in (**a**); solid line: linear fit for the experimental data points.

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
