# Peer review of "Correlative Confocal Raman and Scanning Probe Microscopy in the Ionically Active Particles of LiMn2O4 Cathodes"

_materials, 2019, doi:10.3390/ma12091416_

Round 1
Reviewer 1 Report
This manuscript is a "first draft" that needs to be edited for logic and content.
It should not be published as is.
general:
The paper makes a claim along the lines; we have eliminated A and B, so it must be C.
The claim that the effects that they see are Vegard strain is made by elimination, not by a positive proof by measured data. This is not scientific.
There is no data analysis. There are few numbers in this paper, some are just stated without any proof. There is no error analysis and no error bars on any of the data. One of the main claims is between a 1.2 and 1.3 pmwithout any sense of the reliability of these numbers.
For a paper about ions, there is only one numerical value for ion concentration, line 364, and the data is not from this paper!
There are several long sections that contain equations. Theer equations are not used tofit data, or to inform graphs or a numerical analysis. They should all be deleted. The section of the space charge, line 298, has no refernces, and I am not sure if the authors have shown that this equation is relevant.
Details:
I was not able to list all of the error, there are just too many. Did all of the authos see this manuscript?
Line 66; in what world are 2016 and 2018 "simultaneous"?
Line 83, Raman spectroscopy is not a structural measurement. XRD is a structure measurement. In all cases where Raman spectra are used to determine a specific structural configuration, the relationship is confirmed through another method.
line 101: "lines' not grids
line 148; give the statistics on these measurements.
line 158: intense, not intensive. But adjectives like this one are out of place in scientific writing.
Figure 1c, "KHz" are not the correct units.
line 171: What ARE the ion concentrations? You can use a well characterized sample to test a method, or a well characterized method to test a sample. An unreliable sample and an unreliable method give you nothing.
line 174: how do you measure "inside" a particle with an AFM?
Figure 4 is useless. No trends are observed. No units in e) and h). Also, the curve in b) looks symmetric. What criterion was used to determine asymmetry?
Figure 5 is also unnecessary.
I suggest the authors return to the data. Measure some numbers and do statistical analysis.
Rewrite this paper, if possible, and separate the results from the discussion.
Author Response
Dear Ms. Freya Dong,
please find for your consideration for publication in Materials a revised version of our manuscript “Correlative confocal Raman and scanning probe microscopy in the ionically active particles of LiMn2O4 cathodes” by Denis Alikin, Boris Slautin, Daniele Rosato, Vladimir Shur, Alexander Tselev and Andrei Kholkin.
We have carefully examined reviewers’ comments to our manuscript. We greatly appreciate the reviewers’ efforts in carefully reading the manuscript and providing a constructive criticism, which allowed a significant improvement of its quality. We rigorously addressed comments and suggestions of the referees and revised figures and text accordingly. The changes are highlighted in yellow in the revised version of the manuscript. Some paragraphs have been completely rewritten. To avoid confusions noted by the referees, we added additional description of experimental details and changed the order of the presentation of the results. Our detailed responses to the reviewers’ comments are below.
We hope that after revisions, the manuscript meets the high standards of Materials and is suitable for publication in your reputable journal. We are looking forward to your decision.
On behalf of the authors
Yours truly, Denis Alikin
The paper makes a claim along the lines; we have eliminated A and B, so it must be C.
The claim that the effects that they see are Vegard strain is made by elimination, not by a positive proof by measured data. This is not scientific.
Apparently, a bad way to present the experimental data and the construct the discussion in the original version of the manuscript lead to a confusion noted by the referee. In fact, we do not try to prove existence of the Vegard strain in the ESM measurements because it has been already clearly shown in earlier experiments in many ionic materials by different groups. For example:
1. Lithium manganate-based thin films [1][2]
2. Lithium cobaltite-based thin films [3,4]
3. Amorphous silicon anodes [5][6]
4. Nanocrystalline LiFePO4 [7]
These findings were supported theoretically [8,9]. We as well recently validated the existence of frequency dependence of electrochemical strain microscopy (ESM) response corresponding to Vegard strain in similar commercial battery cathodes [9]. Application of larger voltage biases (30-50 V) in the ESM measurements can create distinct Li particles on the sample surface, which is known from literature [10] and has been validated in our samples too (Figure r1).
Figure r1. Evolution of the topography by application of the 50 V DC voltage pulse in delithiated sample of LiMn2O4 cathode: a) initial grain with marked point of the voltage application, b) hills in topography formed by DC voltage application corresponding to excess and stabilization at the surface of the Li ions (supplemented in cover letter)
Therefore, the existence of the Vegard strain in the presented work was well expected and out of doubt. The aim of the detailed discussion of the eliminated parasitic effects is different. We evaluated possible parasitic contributions to ESM response to make a correct comparison between the confocal Raman microscopy and ESM data. While the existence of the parasitic effect was known in the earlier works, so far there has been no systematic analysis of their contributions as well as demonstrations of effective approaches to eliminate them, like it has been done in the presented work for the case of the local probe-sample electrostatic interaction. The logic of elimination of the original manuscript is not used in the revised version. Instead, the parasitic effects are discussed as parasitic effects contributing together with the Vegard strain effect, whose presence is known.
There is no data analysis. There are few numbers in this paper, some are just stated without any proof. There is no error analysis and no error bars on any of the data. One of the main claims is between a 1.2 and 1.3 pm without any sense of the reliability of these numbers.
We used the full set of calibrations procedures (static calibration of the AFM displacement signal and calibrations based on measurements of the probe vibrational spectra) in full accordance with the procedures established in literature [11,12] to obtain quantitative values of the probe tip displacements. The value obtained for the surface displacement amplitude associated with the Vegard strain—1.3 pm/V—has been obtained by dividing the value of actual surface displacement amplitude of 13 pm by the amplitude of the applied AC voltage of 10 V. It is known that AFM has an extremely high resolution and sensitivity in the vertical direction. Implementation of the long-integration-time lock-in filtering improves the sensitivity even more. The measurement uncertainty level can be estimated from a comparison of the ESM signal from LMO and the electro-mechanically non-active silicon dioxide (Figure r2). It can be estimated to be about 0.25 pm, which corresponds to 0.02 pm/V.
Figure r2. Frequency dependence of the ESM response for LiMn2O4 and the electromechanically non-active silicon dioxide (corresponding to system noise level). The red curve is a fit by equation (5) of the main text of the revised manuscript. Vac = 5 V (supplemented in cover letter)
For a paper about ions, there is only one numerical value for ion concentration, line 364, and the data is not from this paper!
We regret the confusion. This value was calculated from equation (5) of the revised version according to the procedures of ESM data quantification published earlier [9]. The input data for the calculation were taken for the experiments in the presented paper. We have removed the confusing reference and rewritten this part of the manuscript to make it clearer. The numerical values of the Li-ion concentration in the LMO sample are discussed in the forth paragraph of subsection 3.4 of the revised manuscript and compared with the average value deduced from the macroscopic XRD measurements. The ESM and XRD-derived values are in an excellent agreement.
There are several long sections that contain equations. Theer equations are not used tofit data, or to inform graphs or a numerical analysis. They should all be deleted.
We kindly disagree with the reviewer at this point. All the equations in the manuscript are of importance for understanding the discussion; they all are used either for evaluation of the Vegard strain or for evaluations of the parasitic contributions based on the measured data. Equation (5) (previously Equation (1)) is used for the calculations of ion concentration and clarification of the frequency dependence of the ESM response. Equation (1) (previously Equation (2)) is used for calculation of the electrostatic force parasitic contribution in the low-frequency ESM from the values obtained in the high-frequency range. Equation (2) and (3) (previously Equation (3) and (4)) are necessary to understand how to extract surface displacements due to the local Joule heating. They are used for calculations of what fraction of the total dissipating power can impact the electromechanical signal. Equation (4) (previously Equation (5)) is used to fit the I-V curves for a further analysis of the local heating-induced response.
The section of the space charge, line 298, has no refernces, and I am not sure if the authors have shown that this equation is relevant.
We thank the reviewer for the comment. We added references in the text together with an associated discussion. We note that the asymmetry of the I-V curves can be traced to changes of the material conductivity, which is a function of Li-ion concentration (ref [44] in the main text). Therefore, the maps in Figure 3, which are based on the space-charge-limited current model introduced by eq. (4), provide additional information about the Li ion distribution in the particle.
Details:
I was not able to list all of the error, there are just too many. Did all of the authos see this manuscript?
We acknowledge the reviewer for the critical reading of the manuscript. We have carefully examined and proofread the manuscript.
Line 66; in what world are 2016 and 2018 "simultaneous"?
Thank you for pointing this out. We modified the wording in the manuscript.
Line 83, Raman spectroscopy is not a structural measurement. XRD is a structure measurement. In all cases where Raman spectra are used to determine a specific structural configuration, the relationship is confirmed through another method.
We agree with this statement. A complex, comprehensive characterization proving the reliability of the Raman spectroscopy measurements was published earlier in [13,14]. Nevertheless, it must be noted that the focus of the paper is on localized micro- and nanometer-scale measurements of ionic mobility, phase, and lithiation state. Measurements at such a length scale are difficult to perform by XRD. For that reason, we used the confocal Raman microscopy for the correlative measurements in combination with the ESM. The macroscale XRD characterization of identical samples was performed in refs. [30,31] of the main text.
line 101: "lines' not grids
Thank you for pointing this out. We have corrected the mistake in the text of the manuscript.
line 148; give the statistics on these measurements.
In Figure 1b, survey Raman spectra are presented. Raman spectra at each pixel of the mapped area were analyzed, fitted by superpositions of Gaussian functions and plotted as maps of the A1 peak shift and the relative intensities of A1 and Mn3O4-A1g peaks. We added histograms illustrating statistical distributions of the corresponding values of the Raman spectra to the right from both of the maps. For the purpose of a comparison between the ESM and Raman imaging, we choose a typical particle inside the cathode sample with a clear separation of the phases. A more detailed analysis of the Raman data can be found in our earlier publication [14]. We note as well that we made a number of correlative measurements with similar results but do not include all of them in the manuscript because the average ESM response in different particles across the cathode sample has a similar level. For example, please refer to Figure r3.
Figure r3. Complimentary ESM and micro-Raman spectroscopy of an active ionic particle different from that shown in the main text of the manuscript: (a) AFM topography image of the area mapped with the micro-Raman spectroscopy. (c) Distribution of the A1 Raman peak shift corresponding to the local lithiation state. (d) ESM amplitude image (supplemented in cover letter)
line 158: intense, not intensive. But adjectives like this one are out of place in scientific writing.
Thank you for pointing out this misprint. We have changed the word to “strong”.
Figure 1c, "KHz" are not the correct units.
Thank you for pointing this out. We corrected the units to the ‘cm-1’
line 171: What ARE the ion concentrations? You can use a well characterized sample to test a method, or a well characterized method to test a sample. An unreliable sample and an unreliable method give you nothing.
Following your earlier comment, in the revised version, we show the value of the averaged ion concentration obtained based on bulk XRD measurements of the cathode sample—1.15∙1027 m-3—and compare it with values extracted from the ESM measurements that showed that the local concentration across the particle area varies from 0.7∙1027 m-3 to 1.4∙1027 m‑3. The values are in an excellent agreement. It should be also noted that the ESM has been approbated in a wide range of materials. Quantitative ESM measurements, for example, have been made on silicon anodes [16], lithium cobaltite-based films [1,6], Li-ion conductive glass ceramics [17], and on commercial battery materials [9,19].
line 174: how do you measure "inside" a particle with an AFM?
Thank you for pointing this out. We have changed the wording in the corresponding sentences to avoid possible confusions.
Figure 4 is useless. No trends are observed.
We kindly disagree with this statement. First, the current maps in Figure 3 of the revised version (Figure 4 of the original version) indicate that ESM signal is not related to variation of the contact area or an electronic conductivity change. Second, the figure displays a typical I-V curve and distributions of the fitting parameters. They are used further for the calculation of the local heating response. Additionally, as we noted above, the asymmetry of the I-V curves can be traced to changes of the material conductivity, which is a function of Li-ion concentration. Therefore, the maps in Figure 3 provide additional valuable information about the Li ion distribution in the particle and is worth showing in the paper.
No units in e) and h).
Figures 3e and h (original Figure 4) show distributions of the power n in the fitting equation 5 and of the ratio A+/A-, respectively. Both quantities are dimensionless.
Also, the curve in b) looks symmetric.
In fact, it is not. Generally, a perfect asymmetry in not expected based on the difference of the sign and behavior of the injected carriers: electrons vs. holes. Plus, the conductivity is a function of the Li ion concentration which is modified oppositely by application of voltages of opposite polarities.
What criterion was used to determine asymmetry?
The asymmetry is relatively weak, but it can be seen from different values of current at equally valued biases for positive and negative branches of the curve. We characterized the asymmetry by the ratio of the fitting coefficients A+/A- in eq. (4).
Figure 5 is also unnecessary.
We agree that the original Figure 5 is not very important, however, it can help with understanding the frequency behavior of the thermal expansion due to the local heating resulting from the asymmetry of the I-V curves. We moved it from the main text to Appendix.
I suggest the authors return to the data. Measure some numbers and do statistical analysis.
The manuscript is not focused on a quantitative analysis of the ionic concentration across the cathodes. Generally, we did not observe large variation of the ion concentration neither inside the individual particles, nor between different grains of the cathodes. Although, the measurements were made in a number of cathode particles, only particles with clearly seen different phases were chosen for the analysis and presentation. Such a choice is the most useful for understanding how the ESM and Raman data are correlated. In the revised version of the paper, we included additional quantitative and statistical analysis, in particular, in the new Figure 5.
Rewrite this paper, if possible, and separate the results from the discussion.
Thank you for the advice. We revised the structure of the paper, changed the presentation order, and replaced one of the figures. However, we believe that a total separation of result from discussion would make the paper more difficult for understanding. The selected structure of the presentation is important for the step-by-step description of the implemented methodological approach.
1. Yang, S.; Yan, B.; Wu, J.; Lu, L.; Zeng, K. Temperature-Dependent Lithium-Ion Diffusion and Activation Energy of Li 1.2 Co 0.13 Ni 0.13 Mn 0.54 O 2 Thin-Film Cathode at Nanoscale by Using Electrochemical Strain Microscopy. ACS Appl. Mater. Interfaces 2017, 9, 13999–14005.
2. Zeng, K.; Li, T.; Tian, T. In situ study of Li-ions diffusion and deformation in Li-rich cathode materials by using scanning probe microscopy techniques. J. Phys. D. Appl. Phys. 2017, 50, 313001.
3. Balke, N.; Jesse, S.; Morozovska, a N.; Eliseev, E.; Chung, D.W.; Kim, Y.; Adamczyk, L.; García, R.E.; Dudney, N.; Kalinin, S. V Nanoscale mapping of ion diffusion in a lithium-ion battery cathode. Nat. Nanotechnol. 2010, 5, 749–754.
4. Yang, S.; Yan, B.; Li, T.; Zhu, J.; Lu, L.; Zeng, K. In situ studies of lithium-ion diffusion in a lithium-rich thin film cathode by scanning probe microscopy techniques. Phys. Chem. Chem. Phys. 2015, 17, 22235–22242.
5. Balke, N.; Jesse, S.; Kim, Y.; Adamczyk, L.; Tselev, A.; Ivanov, I.N.; Dudney, N.J.; Kalinin, S. V. Real space mapping of Li-ion transport in amorphous Si anodes with nanometer resolution. Nano Lett. 2010, 10, 3420–3425.
6. Simolka, M.; Heim, C.; Friedrich, K.A.; Hiesgen, R. Visualization of Local Ionic Concentration and Diffusion Constants Using a Tailored Electrochemical Strain Microscopy Method. J. Electrochem. Soc. 2019, 166, A5496–A5502.
7. Nataly Chen, Q.; Liu, Y.; Liu, Y.; Xie, S.; Cao, G.; Li, J. Delineating local electromigration for nanoscale probing of lithium ion intercalation and extraction by electrochemical strain microscopy. Appl. Phys. Lett. 2012, 101, 063901.
8. Tselev, A.; Morozovska, A.N.; Udod, A.; Eliseev, E. a; Kalinin, S. V Self-consistent modeling of electrochemical strain microscopy of solid electrolytes. Nanotechnology 2014, 25, 445701.
9. Alikin, D.O.; Romanyuk, K.N.; Slautin, B.N.; Rosato, D.; Shur, V.Y.; Kholkin, A.L. Quantitative characterization of the ionic mobility and concentration in Li-battery cathodes via low frequency electrochemical strain microscopy. Nanoscale 2018, 10, 2503–2511.
10. Arruda, T.M.; Kumar, A.; Kalinin, S. V; Jesse, S. The partially reversible formation of Li-metal particles on a solid Li electrolyte: applications toward nanobatteries. Nanotechnology 2012, 23, 325402.
11. Balke, N.; Jesse, S.; Yu, P.; Carmichael, B.; Kalinin, S. V; Tselev, A. Quantification of surface displacements and electromechanical phenomena via dynamic atomic force microscopy. 2016.
12. Rabe, U. Atomic Force Acoustic Microscopy.
13. Amanieu, H.Y.; Aramfard, M.; Rosato, D.; Batista, L.; Rabe, U.; Lupascu, D.C. Mechanical properties of commercialLixMn2O4cathode under different States of Charge. Acta Mater. 2015, 89, 153–162.
14. Slautin, B.; Alikin, D.; Rosato, D.; Pelegov, D.; Shur, V.; Kholkin, A. Local Study of Lithiation and Degradation Paths in LiMn2O4 Battery Cathodes: Confocal Raman Microscopy Approach. Batteries 2018, 4, 21.
15. Baddour-Hadjean, R.; Pereira-Ramos, J.-P. Raman Microspectrometry Applied to the Study of Electrode Materials for Lithium Batteries. Chem. Rev. 2010, 110, 1278–1319.
16. Jesse, S.; Balke, N.; Eliseev, E.; Tselev, A.; Dudney, N.J.; Morozovska, A.N.; Kalinin, S. V. Direct mapping of ionic transport in a Si anode on the nanoscale: Time domain electrochemical strain spectroscopy study. ACS Nano 2011, 5, 9682–9695.
17. Seol, D.; Park, S.; Varenyk, O. V.; Lee, S.; Lee, H.N.; Morozovska, A.N.; Kim, Y. Determination of ferroelectric contributions to electromechanical response by frequency dependent piezoresponse force microscopy. Sci. Rep. 2016, 6, 30579.
18. Chen, Q.N.; Adler, S.B.; Li, J. Imaging space charge regions in Sm-doped ceria using electrochemical strain microscopy. Appl. Phys. Lett. 2014, 105, 1–5.
19. Luchkin, S.Y.; Romanyuk, K.; Ivanov, M.; Kholkin, A.L. Li transport in fresh and aged LiMn2O4 cathodes via electrochemical strain microscopy. J. Appl. Phys. 2015, 118, 072016.

Reviewer 2 Report
Please see the attached file.

Author Response
Dear Ms. Freya Dong,
please find for your consideration for publication in Materials a revised version of our manuscript “Correlative confocal Raman and scanning probe microscopy in the ionically active particles of LiMn2O4 cathodes” by Denis Alikin, Boris Slautin, Daniele Rosato, Vladimir Shur, Alexander Tselev and Andrei Kholkin.
We have carefully examined reviewers’ comments to our manuscript. We greatly appreciate the reviewers’ efforts in carefully reading the manuscript and providing a constructive criticism, which allowed a significant improvement of its quality. We rigorously addressed comments and suggestions of the referees and revised figures and text accordingly. The changes are highlighted in yellow in the revised version of the manuscript. Some paragraphs have been completely rewritten. To avoid confusions noted by the referees, we added additional description of experimental details and changed the order of the presentation of the results. Our detailed responses to the reviewers’ comments are below.
We hope that after revisions, the manuscript meets the high standards of Materials and is suitable for publication in your reputable journal. We are looking forward to your decision.
On behalf of the authors
Yours truly, Denis Alikin
1. In all Figures and captions, authors mentioned delithiated cathode in the ‘Materials and Methods’ section of the manuscript. However, the explicit explanation and condition for the sample are required for which the complimentary AFM and micro-Raman spectroscopy conducted should be added. For example, ‘3 cycled and delithiated sample’ or ‘pristine’
Thank you for the comment. We added relevant information regarding cycling into the Materials and Methods section of the manuscript.
2. In Figure 2 and its caption, (j) should be corrected to (i).
Thank you for pointing this out. We have corrected this mistake.
3. In ‘materials and methods’, authors mentioned that before the measurements, cathodes were extracted from a fresh, full charged battery. Rather than this, authors should provide a charging profile containing its capacity (mAh g-1 ) and voltage (V). In addition, what ‘fully charged’ stands for should be explicitly explained. For example, the active material, LiMn2O4, was delithiated with 0.1 C-rate and up to 4.5 V (vs. Li/Li+).
Thank you. We added the following sentences in Materials and Methods section:
“Li-battery cathodes (positive electrodes) were investigated in our experiments. The cathode samples consisted of Li0.61Mn2O4 (LMO) active particles embedded in a carbon binder mixed with poly(vinylidene fluoride) (PVDF). The particles have a cubic spinel lattice structure with a nominal lattice parameter of 0.809 nm determined from X-ray diffraction measurements [30,31]. The average concentration of Li ions can be estimated from this data to be about 1.15∙1027 m-3.∙The LMO cathode was prepared and tested in the electrochemical cylindrical cells at Robert Bosch GmbH. It was cycled against the graphite anodes at an 1 C-rate from 2.5 V to 4.2 V 3 times at an 1 C charge rate and an 1 C discharge rate. The cells were opened in an argon-filled glove box. The cathode, separator and anode were uncoiled and separated from each other. Pieces of the cathode on the current collector were cut and washed in a dimethyl carbonate solution in order to remove the electrolyte. After that, samples were taken out of the glove box and dried. Then, the cathodes were thoroughly polished by step-by-step decreasing the polishing abrasive grit size with a finish by the stress-free silica fine polishing. As the last step, the sample surface was treated by Ar plasma to thin and possibly remove the damaged surface layer. A more detailed procedure of sample preparation can found elsewhere [17,18].”
4. In 3.2.2. Local Joule heating, authors assumed that the I-V curve shape can be described by the space-charge-limited (SCLC) model. For that, authors should provide any references of it or rational reason and justification for the assumption.
Thank you for this remark. We provided corresponding references and sentences in the revised text.
“Application of a nominally small bias voltage to an AFM tip induces a quite high local electric field (see Appendix B). The field value is above 10-7 V/m, that is, high enough for electron and hole injection into a sample [45]. The injection of additional charge carriers in the material leads to a space charge limited electronic transport (space-charge-limited current, SCLC) [45–47].”

Round 2
Reviewer 1 Report
This manuscript is much improved. It may be published.